# Study on Comparing the Performance of Fully Automated Container Terminals during the COVID-19 Pandemic

**Bokyung Kim [1], Geunsub Kim [1]** and **Moohong Kang [2],***

1   Port Research Department, Korea Maritime Institute, Busan 49111, Korea; kimb@kmi.re.kr (B.K.); gskim@kmi.re.kr (G.K.)
2   Logistics and Maritime Industry Research Department, Korea Maritime Institute, Busan 49111, Korea
*   Correspondence: mkang@kmi.re.kr; Tel.: +82-51-797-4684

**Abstract:** Major ports worldwide suffered from various problems such as labor shortage, port congestion, and global supply chain disruptions during COVID-19. To ensure stable operations of ports in such scenarios, one option is to adopt fully automated terminals. This study aimed at evaluating the performance of fully automated terminals compared with that of non-fully automated ones during the disrupted logistics due to coronavirus. Four ports that simultaneously operate both fully and non-fully automated terminals were selected. The performance of the target terminals was measured based on quantitative factors: throughput, number of ship arrivals, and berthing time. The results showed that the fully automated container terminals present better operational performance than the non-fully automated terminals. The former achieved large increments in the throughput, small decreases in the berthing time per ship, and increases in the number of ship arrivals. Moreover, there were economic benefits, revealing that the total terminal profit at the fully automated terminals was significantly increased, whereas that at the non-fully automated ones decreased based on berthing times. Therefore, fully automated terminals can be considered as alternatives for responding flexibly and stably during crises such as COVID-19.

**Keywords:** COVID-19 pandemic; supply chain; automated container terminal; port performance; AIS data analysis



## 1. Introduction

The recent COVID-19 outbreak caused major ports worldwide to experience various inefficiencies. Continual labor shortages were caused by coronavirus infections in port and hinterland logistics facility workers. Simultaneously, port functions were paralyzed by the increase in the throughput due to the temporary surge in the demand for consumer goods, which was previously shrinking. Port congestion occurred as ports experienced problems with increased ship waiting times and a lack of storage yards, which rapidly disrupted global supply chains [1,2]. The global supply chain disruptions increased logistics costs including freight rate, and, ultimately, affected global inflation [3].

As such, it is necessary to ensure that ports operate stably to allow global supply chains to flow smoothly under uncertain international conditions. Ports have long been recognized as elements in global supply chains; however, in the current international conditions, uncertainty is spreading, and ports are increasingly being considered as core parts of the supply chain infrastructure [4,5]. This sheds light on how important ports are for a stable operation of the entire global supply chain. Specifically, ports act as the nexus in global supply chains, and ensuring their stability is crucial to enable them to respond flexibly and rapidly during crises [6].

Fully automated terminals may be one approach for ensuring the stability of port operations. Currently, in ports worldwide, most container terminals are operated in a semi-automated or manual manner, with a few exceptions. Only approximately 4% of container

terminals globally have introduced automation technology of any type, whether full or semi-automation [7]. Therefore, ports have a limited ability to maintain stable operations during international disasters such as the COVID-19 outbreak. However, in fully automated terminals, all processes operate without people; therefore, relatively flexible operations are expected to be possible during changes in international conditions [8].

Container terminal automation technology was first introduced at the ECT Delta terminal in Rotterdam, Netherlands in 1993, and at the time, it was recognized as a new and innovative industrial field [9]. Subsequently, full automation, which included unmanned quayside operations, began at the RWG and AMPT terminals at the Rotterdam port in 2015, and it was introduced at the Qingdao and Yangshan port in Shanghai, China, and continues to be in operation there. However, it has not been long since fully automated ports were introduced worldwide, and their effect is gradually increasing with the improvement in their technological reliability. As such, their actual operational performance has been small [10]. Even currently, highly productive container terminals operate in a manual or semi-automated manner. In addition, since the introduction of fully automated terminals, there has been no precedent for the type of severe port congestion caused by a crisis such as COVID-19, and it has not been possible to verify whether actual fully automated terminals operate effectively.

The objective of this study was to evaluate the extent to which fully automated terminals perform better than non-fully automated ones in the state of disrupted logistics due to the COVID-19 pandemic. Specifically, this study aimed to verify whether a fully automated terminal can operate stably, using actual port data to compare operational performance. This study is distinct from previous ones in that it examined the performance of fully automated container terminals during the COVID-19 pandemic and considered them as alternatives that can ensure port stability and flexibility. In addition, it used actual data to comparatively analyze the performance of ports that have fully automated terminals, based on which its conclusions were derived. It was empirically analyzed whether it is possible to secure the stability and flexibility of a fully automated terminal using actual performance data rather than methods such as simulation and scenario analysis. Through this, it is possible to evaluate whether the stability of a port is important and in crisis situations such as COVID-19, and whether a fully automated terminal can be an alternative.

Section 2 addresses the importance of ports and the operation effect of fully automated terminals during the COVID-19 crisis through a literature review of previous studies, and proposes a differentiation of the performance analysis using empirical data. Section 3 introduces the analysis targets and analysis methods, and Section 4 presents the analysis results and verifies the operation performance of fully automated terminals. Finally, in the conclusion, it is suggested that a fully automated terminal is effective in terms of stability amid the changing external conditions of ports.

## 2. Literature Review

After the COVID-19 outbreak in 2020, studies were conducted to examine its effect on the shipping and ports industries as well as supply chains. In most studies, COVID-19 showed a considerable negative effect on the maritime transport and supply chains [1,11,12]. In particular, it was found that the logistics delays caused by port congestion were a major factor in the global supply chain problems [6].

The problems caused by the port delays during COVID-19 include a reduced number of ports of call for shipping lines, inland transport delays, insufficient equipment/facilities, labor shortage, and surges of utilization rates in storage facilities [12–14]. Studies using actual data confirmed the aforementioned individual threat factors and their effects [15–17]. For example, Gui et al. [2] analyzed port congestion factors caused by COVID-19 based on hinterland transportation networks, labor shortages, and equipment/facility shortages, and evaluated the priority of the risk level of each factor. Major research organizations of the world also support this conclusion with their own analyses. The port congestion index

reached a record highest value of 37.3% in October 2021 [18], and global liner schedule reliability became a record low, 33.6%, during the same period owing to port congestion [19].

Such disruptions in supply chains significantly reduce the efficiency of all global supply chains and increase logistics costs, such as freight rates, which inevitably increase the inflationary pressure. The United Nations Conference on Trade and Development (UNCTAD) [3] found that the surge in the freight rates resulted in a 10.6% increase in the global import costs and a 1.5% increase in the cumulative consumer prices (2020–2023). The Organization for Economic Cooperation and Development (OECD) [7] and the European Central Bank [20] also performed a similar analysis and found that increase in logistics costs such as freight rates affected the consumer price increase of the following year.

Recent studies have argued that an approach that ports must adopt after the COVID-19 pandemic is to ensure sustainable resilience to allow them to respond to crises [6,21]. Specifically, there is a need to maintain resilience, flexibility, and stability while pursuing the same level of efficiency as in the past. As an approach to achieve this, Alamoush et al. [12] emphasized active used of smart (digital) technology. Notteboom et al. [22] reviewed port operational and financial data, and found that automated (digitized) ports are playing important roles in improving the resilience and the ability to respond to crises during COVID-19.

Therefore, fully automated terminals can be considered as alternatives for responding flexibly during crises such as the COVID-19 pandemic. This is because they can continuously operate without interruption during port worker shortages and their performance remains constant [7]. This is in line with the argument proposed by Rodrigue and Notteboom [23]. Specifically, terminal automation seeks to provide stability, predictability, and consistency of operational performance, instead of improving the operational efficiency, i.e., productivity. Therefore, it aims to reduce the quantity and length of operational interruptions and ensure continuous operations.

In general, introducing automation technology at terminals has the effects of improving productivity, reducing costs, achieving environmental sustainability, and service-level improvements such as in speed. Kon et al. [9] analyzed papers published between 1993 and 2019, and found that many research results confirm the above, and they stated that the frequency of automation technology usage will expand further in the future. Specifically, Park and An [24] found that fully automated terminals that are operated without people have the effect of reducing costs by 37%, reducing carbon emissions by 50%, and increasing productivity by 40%. However, Nam and Ha [25] mentioned that unmanned automation systems are not cost-effective compared with traditional systems; these study results were derived by experiments based on modeling and simulations, which may be different from actual operations. Wang et al. [26] observed the relationship between the terminal type, strategy, performance, and adoption of automation in 20 major container terminals of the world. They found that automation is being introduced for pursuing efficiency or as a marketing strategy measure in accordance with the strategy of the terminal. In addition, they claimed that automation may provide the benefit of achieving productivity and stability as well as reliability. However, Wang et al. [26] examined terminals that introduced automation before 2014, and these terminals may be different than the unmanned fully automated terminals that are discussed in this study.

The results of reviewing previous studies show that ports play pivotal roles in global supply chains, and automated container terminals have various environmental and economic effects. However, few studies have focused exclusively on unmanned fully automated terminals, and investigations discussing the performance of fully automated terminals in particular are lacking. This is because most studies on the effects of automated terminals do not distinguish between terminals that operate with unmanned full automation and other types of automation [8]. In addition, the majority of previous research verified the effects of introducing automated terminals in regard to technical aspects such as terminal layout types and the allocation of equipment and space, as in the case of studies by Aisha et al. [27], Qin et al. [28], and Wang et al. [29].

Studies on whether fully automated terminals were operated effectively during COVID-19 have also been lacking. They have been limited to studies by Guerrero et al. [30] on changes in port networks before and after the COVID-19 pandemic (2019 and 2020) and by Wang et al. [31] analyzing ship pattern changes in ports before and after the COVID-19 outbreak in 2020. In this context, the World Bank [5] also noted that there is no reliable performance indicator for comparing port operational performance even though the importance of ports has become evident since the COVID-19 outbreak.

This study differentiated recently introduced fully automated container terminals from other terminals, which has not been attempted before, and it analyzed the operational performance of actual container terminals instead of performing a technology-based, simulation-oriented analysis. In addition, this study distinguishes itself from previous studies in that it analyzed the operational performance of ports with fully automated terminals before and after the COVID-19 outbreak. Considering the above, this study aimed to verify whether fully automated terminals operated stably during COVID-19 by comparing the actual operational performance of fully automated terminals to that of non-fully automates ones.

## 3. Materials and Methods

### 3.1. Analysis Targets

There were 55 automated terminals in the world as of the year 2020 [23]. Based on this fact, in this study, the analysis unit is defined as a container terminal. A fully automated terminal that is discussed in this paper refers to a terminal in which all processes operate without people, the quayside work is mainly operated by remote control, and the yard is operated by ARMG (automated rail-mounted gantries) and AGV (automated guided vehicles) [8]. In contrast, non-fully automated terminals are terminals that operate in a semi-automated or manual manner. They are defined as terminals in which the yard and transport equipment are manned or unmanned and the quayside work is performed by people (refer to Table 1). Automated terminals are variously mentioned in many papers, such as in the context of fully automated, automated, semi-automated, and manual [8,23,32], but in this study, they were divided into only two types: fully automated terminals and non-fully automated terminals.

**Table 1.** Analyzed ports and terminals.

| Category | Fully Automated | | Non-Fully Automated | |
|---|---|---|---|---|
| | **Port** | **Terminal** | **Port** | **Terminal** |
| Port | Rotterdam | RWG, APMT | Rotterdam | ECT Delta, Euromax |
| | LA/LB | LBCT, TraPac | LA/LB | TTI(Pier T), SSA(Pier A), APMT |
| | Qingdao | QQCTN | Qingdao | QQCT, QQCTU |
| | Shanghai | Yangshan Port Phase 4 | Shanghai | Yangshan Port Phases 1–2, Phase 3 |
| Total | | 6 terminals | | 9 terminals |

The analyzed ports were limited to those where fully and non-fully automated terminals are operated simultaneously. Referring to previous studies, a total of four ports that operate fully automated terminals were selected, which were those at the Rotterdam port in the Netherlands, LA/LB (Los Angeles/Long Beach) port in the United States, and Qingdao and Yangshan ports in Shanghai, China. To compare the performance of terminals under the same conditions, this study selected non-fully automated terminals that were also located in the four selected ports.

### 3.2. Performance Factors

To compare the performance of container terminals, it is necessary to select factors that measure performance. First, throughput is the most fundamental and general performance measurement factor that shows the growth and market share of a port [4]. In addition, in regard to the performance related to port operations, it is possible to evaluate quantitative

performance using factors such as arrival of ships, loading/unloading time, turnaround time, and berth and yard usage rates [3]. As a typical example, the container port performance index published by the World Bank [5] defines and measures performance factors based on the turnaround time of a port. In addition to quantitative factors, port performance may consider qualitative factors such as corporate social responsibility activities, provision of logistics services, and safety and security levels.

However, this study considered only quantitative factors, because of which it is possible to compare performance of individual container terminals during operational processes according to whether they are automated. Specifically, this study measured the performance of the target terminals based on the throughput, number of ship arrivals, and berthing time (work time–loading/unloading time), which are commonly used quantitative performance factors. Turnaround time in a port and waiting time can also be regarded as performance related to port operation, but it is not considered in this analysis because it is difficult to judge that the operation of the automated terminal has a direct impact. If a fully automated terminal is operated stably, the loading/unloading work is performed efficiently, and the berthing times of ships are reduced. This can increase the annual number of ship arrivals, and ultimately increase the handled throughput of an entire terminal.

### 3.3. Analysis Methods

### 3.3.1. Throughput

For the throughput, the total volume that was handled at the individual terminals annually was set in twenty-foot equivalent unit (TEU), and the terminal performance was compared for 2019 and 2020, i.e., before and after the COVID-19 outbreak. For the performance, this study preferred to use the throughput performance for each terminal that is presented in the annual reports of the Drewry Maritime Research, "Global Container Terminal Operators [33,34]". The data contain the throughput of each terminal for which there is ownership or shareholding, based on the terminal operator in each port. There are cases where different throughput data are presented for the same terminal. It is noted that there may be some discrepancies because the data are based on figures reported by terminal operators. Therefore, in such cases, this study preferred to use the throughput presented by the terminal operator with a large stake.

### 3.3.2. Berthing Times and Number of Ship Arrivals

The number of ship arrivals and berthing times were measured using an automated identification system (AIS). An AIS is a device that was developed to resolve security problems such as port control and collisions by determining when and where a ship passed through and sharing this information. Since July 2002, new ships of over 300 tons have been required to install an AIS. Moreover, it is stipulated that an AIS must provide location information at intervals of at least two seconds and at most three minutes, although the transmission interval varies according to the target ship type.

An AIS generally provides three types of information: static, dynamic, and voyage. Of these, static information includes information that does not change, including the dimensions of the ship, and voyage information includes information that occurs during each voyage of the ship, such as the departure and arrival locations. Dynamic information includes the position information of the ship, such as the IMO (International Maritime Organization) number, current latitude and longitude, time, and speed of the ship. This AIS information is used in various research because it can confirm the ship movement patterns, movement times, and arrival ports of target seas and ports worldwide. In representative previous studies related to the present one, Leonardo et al. [35] and Vershuur et al. [11] used global AIS data before and after the COVID-19 pandemic to estimate trade flows (trade volume) and ship movement, focusing on worldwide ports. In contrast, Ito et al. [16] used AIS data to analyze the extent of COVID-19 infections in ports receiving cruise ships as a part of an empirical analysis of the correlation between cruise ship operations and COVID-

19 worldwide. In addition, Guerrero et al. [30] used AIS data to analyze port network structures and make comparisons before and after the COVID-19 pandemic.

Among the various types of AIS information, this study used dynamic information such as ship position and time information to analyze the number of ship arrivals and times at each target terminal. First, berthing time (BT) refers to the time spent in the moorage, and as shown in Figure 1a, the difference between the time of entry ($m_{in}$) and the time of departure ($m_{out}$) of ships in the set moorage was calculated by connecting the ship positions of the AIS on the map. In addition, the waiting time (WT) was calculated as the waiting time at the anchorage, and the total time in a port (TT), previously referred to as turnaround time, was calculated as the time difference between entering and leaving the harbor limit. Finally, the moving time (MT) was calculated as the total time in a port (TT) minus the berthing time (BT) and waiting time (WT). However, some ports do not have information on anchorage, such as Shanghai Port, Rotterdam Port, and LA Port; in this study, waiting and moving times (WT, MT) were calculated and analyzed together. Figure 1b shows the actual moorage for each terminal and the harbor limit in the LA Port, and entry time and departure time were identified by AIS to calculate the time in a port and berthing time of the ships. The ship arrival times were calculated by adding the number of ships (N) that entered and exited the moorage, which is expressed as follows:

$$BT = \sum_{i}^{N} m_{out,\,i} \;-\; m_{in,\,i} \tag{1}$$

$$WT = \sum_{i}^{N} a_{out,\,i} \;-\; a_{in,\,i} \tag{2}$$

$$TT = \sum_{i}^{N} h_{out,\,i} \;-\; h_{in,\,i} \tag{3}$$

$$MT = TT - BT - WT \tag{4}$$

where i is the ship number, N is the total number of ships, $m_{out,\,i}$ is the time when the ith ship leaves the moorage, and $m_{in,\,i}$ is the time when the ith ship enters the moorage, $a_{out,\,i}$ is the time when the ith ship leaves the anchorage, and $a_{in,\,i}$ is the time when the ith ship enters the anchorage, $h_{out,\,i}$ is the time when the ith ship leaves the port, and $h_{in,\,i}$ is the time when the ith ship enters the port.

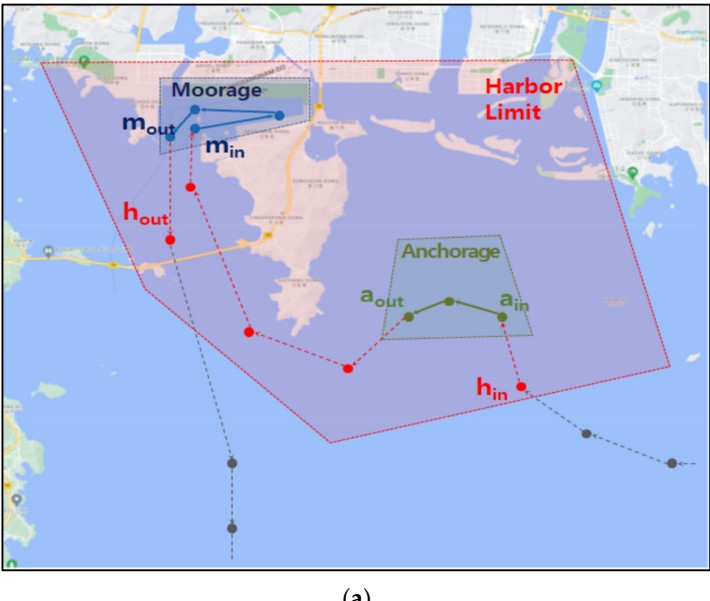

(**a**)

**Figure 1.** *Cont.*

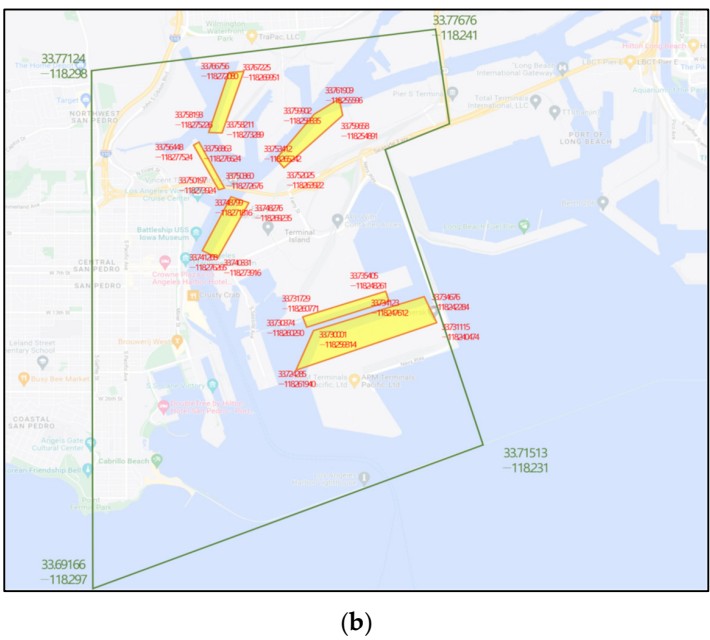

(**b**)

**Figure 1.** (**a**) Identification of route of ship movement using AIS; (**b**) example of setting moorage area (LA port).

To perform this analysis, R and Java programming languages were used together. Specifically, R was used to process the AIS data (sorting and filtering) and Java was used to analyze the ship movement paths and calculate the berthing times and the number of arrivals. The AIS data used in this study were purchased from IHS Markit. However, commercial AIS data vendors such as IHS Markit and VesselsValue only provide AIS data that are received approximately once per hour from each ship. This receiving period is owing to reduced data reception rates, filtering caused by data errors, and improvements in data management and transmission efficiency. As such, this study may be limited by the reduced accuracy caused by this long receiving period.

## 4. Results and Discussion

The results of the analysis conducted in this study showed that the container terminals that were operating in a fully automated manner exhibited better operational performance than the non-fully automated terminals in the same port. First, based on the comparison of the handled throughput summarized in Table 2, there was a trend of increase at all six fully automated terminals between 2019 and 2020, with a large overall increase of 30.18%. With the exception of the AMPT terminal of Rotterdam, most ports showed large double-digit increments. However, in the case of the non-fully automated terminals, the annual throughput was decreased by an average of 1.91%. Certain terminals showed some increase; however, the rate of increase was low, except for the TTI terminal of the LA port. Non-parametric verification using the Mann–Whitney U test was conducted to statistically verify the differences in the result values between a complete automation terminal and a non-complete automation terminal. As a result, the average throughput of fully automated terminals and non-fully automated terminals was found to be significantly different at $p = 0.0016$ ($p < 0.05$).

**Table 2.** Throughput performance comparison.

| Category | Port | Terminal | Throughput (1000 TEU) | | |
|---|---|---|---|---|---|
| | | | **2019** | **2020** | **Change (%)** |
| Fully Automated | Rotterdam | RWG | 1921 | 2228 | 15.98 |
| | | APMT | 2323 | 2421 | 4.22 |
| | LA/LB | LBCT | 1159 | 1911 | 64.88 |
| | | Trapac | 790 | 1075 | 36.08 |
| | Qingdao | QQCTN | 1286 | 1690 | 31.42 |
| | Shanghai | Yangshan Phase 4 | 3271 | 4204 | 28.52 |
| | | Average | 1792 | 2255 | 30.18 |
| Non-fully Automated | Rotterdam | ECT Delta | 5300 | 5100 | −3.77 |
| | | Euromax | 2793 | 2455 | −12.1 |
| | LA/LB | TTI (Pier T) | 2100 | 2400 | 14.29 |
| | | SSA (Pier A) | 875 | 812 | −7.2 |
| | | APMT | 2564 | 2284 | −10.92 |
| | Qingdao | QQCT | 8926 | 9103 | 1.98 |
| | | QQCTU | 6348 | 6782 | 6.84 |
| | Shanghai | Yangshan Phases 1, 2 | 8936 | 8672 | −2.95 |
| | | Yangshan Phase 3 | 7601 | 7346 | −3.35 |
| | | Average | 5049 | 4995 | −1.91 |
| Mann–Whitney U test | | | U = 2.00, $p$ = 0.0016 ($p$ < 0.05) | | |

Based on the berthing time results (Table 3), it can be seen that the fully automated terminals showed a small average increase of 4.59%, with the average berthing time per ship increasing from 38.27 h in 2019 to 39.46 h in 2020. In contrast, the non-fully automated terminals showed an average increase of 16.23%, increasing from 28.67 h in 2019 to 35.88 h in 2020, which is a much larger increase that that for the fully automated terminals. It is noticeable that the rate of increase in the average ship berthing times was low at the fully automated terminals, even though the annual throughput significantly increased. In comparison, the non-fully automated terminals presented a high level of increase in the average ship berthing times, even though the throughput was decreased on average. Therefore, it can be inferred that the fully automated terminals are highly stable in regard to loading and unloading works. As a result, the average values of fully automated terminals and non-fully automated terminals were found to be significantly different at $p$ = 0.0879 ($p$ < 0.1). Moving time, waiting time, and total time in the port derived together also showed similar results to berthing time, but unlike total time in port, moving and waiting time were not statistically significant.

Between 2019 and 2020, the number of ship arrivals increased by an average of 21.8% at the fully automated terminals, whereas it is decreased by an average of 3.05% at the non-fully automated terminals. This can be interpreted as showing that fully automated terminals can maintain stable berthing times and allow a relatively large number of ships to arrive, regardless of changes in the external environment, such as COVID-19. In order to examine the difference between the berthing time and the number of arrivals, verification was performed in the same manner as above. As a result, the average values of fully automated terminals and non-fully automated terminals were found to be significantly different at $p$ = 0.0496 ($p$ < 0.05).

**Table 3.** Berthing times and number of ship arrivals performance comparison.

| Category | Port | Terminal | Berthing Time (Hour) | | | Moving and Waiting Time (Hour) | | | Total Time in Port (Hour) | | | Number of Ship Arrivals (Ships) | | |
|---|---|---|---|---|---|---|---|---|---|---|---|---|---|---|
| | | | 2019 | 2020 | Change (%) | 2019 | 2020 | Change (%) | 2019 | 2020 | Change (%) | 2019 | 2020 | Change (%) |
| Fully Automated | Rotterdam | RWG | 10.91 | 12.84 | 17.64 | 50.33 | 49.69 | −1.27 | 61.24 | 62.53 | 2.11 | 1979 | 2037 | 2.93 |
| | | APMT | 12.78 | 12.88 | 0.76 | 33.27 | 35.8 | 7.60 | 46.05 | 48.68 | 5.71 | 1239 | 1080 | −12.83 |
| | LA/LB | LBCT | 80.69 | 78.11 | −3.19 | 6.98 | 8.45 | 21.06 | 87.67 | 86.56 | −1.27 | 106 | 125 | 17.92 |
| | | Trapac | 87.34 | 94.35 | 8.03 | 9.52 | 5.26 | −44.75 | 96.86 | 99.61 | 2.84 | 75 | 100 | 33.33 |
| | Qingdao | QQCTN | 16.23 | 16.95 | 4.47 | 2.83 | 2.46 | −13.07 | 19.06 | 19.41 | 1.84 | 297 | 494 | 66.33 |
| | Shanghai | Yangshan Phase 4 | 21.65 | 21.62 | −0.16 | 1.69 | 2.37 | 40.24 | 23.34 | 23.99 | 2.78 | 827 | 1018 | 23.10 |
| | | Average | 38.27 | 39.46 | 4.59 | 17.44 | 17.34 | −0.56 | 55.70 | 56.80 | 1.96 | 754 | 809 | 21.80 |
| Non-fully Automated | Rotterdam | ECT Delta | 16.27 | 18.72 | 15.04 | 36.24 | 35.65 | −1.63 | 52.51 | 54.37 | 3.54 | 3936 | 3631 | −7.75 |
| | | Euromax | 12.97 | 12.60 | −2.88 | 46.68 | 49.65 | 6.36 | 59.65 | 62.25 | 4.36 | 2040 | 1879 | −7.89 |
| | LA/LB | TTI (Pier T) | 57.13 | 74.01 | 29.55 | 1.28 | 3.14 | 145.31 | 58.41 | 77.15 | 32.08 | 301 | 255 | −15.28 |
| | | SSA (Pier A) | 32.17 | 35.00 | 8.79 | 6.01 | 10.99 | 82.86 | 38.18 | 45.99 | 20.46 | 245 | 297 | 21.22 |
| | | APMT | 72.07 | 107.10 | 48.59 | 3.41 | 1.89 | −44.57 | 75.48 | 108.99 | 44.40 | 232 | 199 | −14.22 |
| | Qingdao | QQCT | 14.08 | 15.41 | 9.46 | 2.25 | 1.76 | −21.78 | 16.33 | 17.17 | 5.14 | 3146 | 3217 | 2.26 |
| | | QQCTU | 17.41 | 17.94 | 3.06 | 2.5 | 3.24 | 29.60 | 19.91 | 21.18 | 6.38 | 1948 | 2148 | 10.27 |
| | Shanghai | Yangshan Phases 1, 2 | 17.98 | 21.64 | 20.35 | 1.3 | 2.82 | 116.92 | 19.28 | 24.46 | 26.87 | 1847 | 1707 | −7.58 |
| | | Yangshan Phase 3 | 17.97 | 20.50 | 14.09 | 1.23 | 1.87 | 52.03 | 19.2 | 22.37 | 16.51 | 1495 | 1368 | −8.49 |
| | | Average | 28.67 | 35.88 | 16.23 | 11.21 | 12.33 | 10.02 | 39.88 | 48.21 | 20.89 | 1688 | 1633 | −3.05 |
| Mann–Whitney U test | | | U = 12.00, *p* = 0.0879 * | | | U = 18.00, *p* = 0.3277 | | | U = 3.00, *p* = 0.0028 ** | | | U = 10.00, *p* = 0.0496 ** | | |

** *p* < 0.05, * *p* < 0.1.

The stable operational performance of fully automated terminals can have significant economic effects on shipping lines and terminals. First, when fully automated terminals are used, the rate of increase in ship berthing times is relatively low, and the ship operating costs of shipping lines can be reduced. Here, in consideration of the profit aspect of the terminal, only the berthing time corresponding to working directly affected by the fully automated terminal was used to examine the effect. The daily ship operating cost was calculated for the average sizes of all ships berthing each terminal, and it was used with the average berthing time and number of ship arrivals in Table 3 to calculate the total ship operating cost. The average sizes of all ships are the average values of the sizes of all ships entering each terminal during the year, and the results derived from the AIS analysis were used. For the daily ship operating costs, this study used the ship cost values for each container ship size obtained by Drewry Maritime Research, "Ship Operating Cost Annual Review and Forecast [36]". The cost was set as the sum of the capital cost and the operational cost (e.g., seafarers, repairs, and maintenance cost), excluding the fuel cost. It was assumed that fuel was not consumed during berthing time; therefore, the fuel cost was excluded from the total ship operating cost calculation.

$$TC = (DC \div 24 \times BT) \times N \tag{5}$$

where TC is the total ship operating cost for the year, DC is the average daily ship operating cost, BT is the average berthing time for one ship, and N is the total number of ship arrivals for the year.

Based on the results of the calculations as summarized in Table 4, at the fully automated terminals, the ship operating cost for all arriving ships increased by approximately $3.82 million owing to the increase in the berthing times (4.59%). However, at the non-fully automated terminals, the ship cost for all arriving ships was increased by $11.53 million (22%) owing to the increase in the berthing times (16.23%). Therefore, the use of fully automated terminals may be effective in reducing the costs of shipping lines by the reduction in the ship operating cost, and it may allow fleet operations to be performed much more efficiently.

**Table 4.** Economic effects of berthing time.

| Category | Average Ship Size (TEU) | | Ship Operating Cost ($/Day) | | Ship Operating Cost per Ship during Average Berthing Times ($) | | Total Annual Ship Operating Cost ($1000) | | Change ($1000, %) | |
|---|---|---|---|---|---|---|---|---|---|---|
| | 2019 | 2020 | 2019 | 2020 | 2019 | 2020 | 2019 | 2020 | | |
| Fully Automated | 6262 | 6636 | 25,344 | 25,780 | 40,410 | 42,385 | 30,463 | 34,290 | 3827 | 12.6 |
| Non-fully Automated | 6831 | 6944 | 25,987 | 26,180 | 31,046 | 39,138 | 52,399 | 63,930 | 11,532 | 22.0 |

Remark: These results are average or total values of six fully automated terminals and nine non-fully automated terminals.

The increase in the throughput handled by fully automated terminals can increase the profit of a terminal (Table 5). At container terminals, throughput is a major factor generating sales, which takes the form of handling charges per TEU. The effects of the profit from the handling charges can be calculated by applying the handling charges of the terminal to the throughput (Table 2). However, the handling charges at an individual terminal are determined by contracts between shipping lines and terminals, and the exact handling charges are not made public; therefore, it is difficult to obtain the exact handling charges. This study used the same handling charges that were found by shipping lines for 20-foot containers in 2021 for each port. Here, $245 was used for the Rotterdam port, $300 for the LA/LB port, $75 for the Qingdao port, and $65 for the Shanghai ports.

**Table 5.** Economic effects of annual container handling throughput.

| Category | Total Handling Charge Income ($1 Million) | | Change ($1 Million, %) | |
|---|---|---|---|---|
| | 2019 | 2020 | | |
| Fully Automated | 1934 | 2435 | 501 | 25.9% |
| Non-fully Automated | 5865 | 5732 | −133 | −2.3% |

Remark: These results are the total income of six fully automated terminals and nine non-fully automated terminals.

The calculation results showed that the total terminal profit at the fully automated terminals increased by $501 million (25.9%) between 2019 and 2020, whereas it decreased by $133 million (−2.3%) at the non-fully automated terminals. The fully automated terminals that are currently in operation are smaller than the non-fully automated terminals in terms of the size and scale of the facilities. The container throughput of the former is unavoidably smaller than that of the latter; however, it was observed that the profits increased with the increase in the handling throughput owing to the stable operations.

In addition to the aforementioned effects, shipping lines can invest in and manage ships more efficiently. Moreover, the frequency of connections between ports and related industries (e.g., tug and pilot services, fuel supply, and mooring services) and inland transport increase as the berth occupancy ratio increases. Therefore, more economic effects than those mentioned above may be achieved in various areas.

## 5. Conclusions

This study used terminal performance factor data to conduct an empirical analysis of the actual operational performance of fully automated terminals that are operated without people compared with that of non-fully automated terminals during the COVID-19 pandemic. The data were obtained from related literature and AISs, and the analysis results showed that the fully automated terminals presented better performance than the non-fully automated terminals for all factors, including the handling throughput, berthing time, and number of ship arrivals. Regarding the effects of such operational performance, ship operating costs can be decreased owing to the reduction in the berthing time and terminal profits are expected to be increased by the increase in the handling throughput.

After COVID-19, global supply chains must be able to respond more stably and flexibly than earlier, and ensuring the stability of ports is crucial because they are at the center of global supply chains. In this respect, fully automated terminals can act as important elements in ensuring port competitiveness after the COVID-19 pandemic. Similar to the results of this study, Xu et al. [17] stated that fully automated terminals could become an option for response plans after the COVID-19 pandemic. Moreover, they suggested that rapid and stable services will be possible in the future, owing to the increase in port throughput and ultra-large container ship arrivals. In particular, the effects of crises such as COVID-19 are larger at regional hub ports and transshipment ports [30]. Therefore, the operating of fully automated terminal, which ensures stability, could become a more important competitive factor at transshipment ports. In addition to the high efficiency shown by fully automated terminals during the COVID-19 pandemic, it was determined that they can significantly alleviate various port operational risks that may be caused by human errors, labor strikes, demographic changes, and climate change. An International Transport Forum/OECD study [7] also suggested that fully automated terminals may become alternatives that can reduce the societal costs due to labor costs and conflicts between labor and management. Previous studies emphasized the effects of fully automated terminals in terms of efficiency (productivity) and eco-friendliness [24], but this study's results revealed that stability brings the greatest merit. Stability effects were only mentioned in terms of qualitative aspects, or theoretical plausibility in previous studies [22,23,32], while this study empirically verified the effects. While ports may consider a fully automated terminal for different reasons [26], stability is a common benefit for all, which should be reflected in port policies that are to be revised or newly made.

As for the limitations of this study, the port operational performance analysis period during COVID-19 was limited to the year 2020. In addition, the selected terminals were limited to ports that operate fully automated terminals, instead of terminals worldwide. In future studies, better results and suggestions can be obtained by overcoming these limitations, specifically categorizing target ports by region and automation level, and also by including the current time in the comparison period, as the COVID-19 pandemic is ongoing. In addition, in the analysis, the performance indicators were limited to three factors related to loading and unloading works. In future studies, the effects of fully automated terminals can be analyzed in more detail by defining and comparing their performance using various performance indicators.

**Author Contributions:** Conceptualization, G.K.; data curation, B.K. and M.K.; formal analysis, B.K. and G.K.; software, M.K.; writing—original draft, B.K., G.K. and M.K.; writing—review and editing, B.K., G.K. and M.K. All authors have read and agreed to the published version of the manuscript.

**Funding:** This research was supported by Korea Institute of Marine Science & Technology Promotion (KMIST) funded by the Ministry of Oceans and Fisheries (Project No.: 20220573).

**Institutional Review Board Statement:** Not applicable.

**Informed Consent Statement:** Not applicable.

**Data Availability Statement:** Not applicable.

**Conflicts of Interest:** The authors declare no conflict of interest.

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
