# Peer review of "Study on Comparing the Performance of Fully Automated Container Terminals during the COVID-19 Pandemic"

_sustainability, doi:10.3390/su14159415_

Round 1

Reviewer 1 Report

A comparative study on the performance of fully automated container terminals during a coronavirus pandemic gives valuable insight into the perspective of resilience building in a key part of global logistic chains. 

The manuscript is well prepared - it fulfilled all crucial requirements of good academic writing. Congratulations!

I only have one technical remark - in several references (e.g. 4, 6, 13, 28, 33), authors' surnames and names have been swapped. Please verify all references in this respect.

Author Response

We appreciate your kind review.

As you suggested, the revised references were added to the manuscript.

Reviewer 2 Report

The work seems very interesting to me, however I consider that non-parametric statistical tools should be used to validate if there is significance between the differences in performance for fully automated and non-fully automated

Author Response

We appreciate your kind review.

As you suggested, the verification results were added to the manuscript.

Reviewer 3 Report

Short overview

Container terminals automation in ports

Container terminals (CTs) automation in ports is a full or partial substitution of terminal operations through automated equipment and processes. The most common definition classifies CT as follows

- A automated CT is when the container yard (CY) and terminal transport between the quay and the CY are automated. This implies that a container is handled automatically from the seaside to the CY and or vice-verse.

- A semi-automated CT only involves an automated staking yard.

In my opinion, this manuscript try to develop investigation related to ''Study on Comparing Performance of Automated and Semi-Automated Container Terminals''.

CTs automation have been a very popular topic to study very often the automation technologies applied to the Quay Cranes, Yard Cranes, Terminal transport systems (Automated Guided Vehicles - AGV; Straddle Carriers; Shuttle Carriers; Automated Lifting Vehicles - ALV; Intelligent Guided Vehicles) and Main Gate.

Total time that ships stay in a port

Total time that ships stay in a port consists of

1.      Waiting time of ships when ships stay at anchor area.

2.      Berthing/unberthing time of ships, i.e. ships move from anchor area to allocated berth and then ship berthing/docking/mooring at the berth. After the loading/unloading stage of ships, ships are unberthing/undocking/unmooring and leave port. It is very often a constant time.

3.      Ship service time at a berth, i.e. loading/unloading stage of ships.

General overview

The focus of this manuscript is not clear. My opinion is to focus on the ''Study on Comparing Performance of Automated and Semi-Automated Container Terminals''. So, description of the automated CT and semi-automated CT and effects of a comparison need to be systematically introduced. In these aspects, deeper analyses are necessary instead of just introducing terminologies. Then, other issues (for example, COVID-19) need to be analyzed from the CT performance point of view. It seems main CT performances are

- Yearly throughput in TEUs.

- Berthing time of ships which is not clearly stated. Please explain it according to the above mentioned.

- Number of ship arrivals.

- Total Annual Ship Operating Cost at Berth - very simple calculation.

- Total handling Charge Income of CTs.

According to aforementioned this manuscript needs major modifications before the acceptance as follows in the Specific comments.

Specific comments

1.    The title of the manuscript should be changed – one alternative has been proposed.

2.    CTs automation should be clearly defined including Automated CTs and Semi-automated CTs.

3.    Some very important papers are missing such as (Nam and Ha (https://ascelibrary.org/doi/abs/10.1061/(ASCE)0733-950X(2001)127:3(171); Orive et al. (doi:10.3390/logistics4010003)) among others.

4.    Keywords of the manuscript should be changed related to ''Study on Comparing Performance of Automated and Semi-Automated Container Terminals''.

5.    THE SERVICE TIME OF SHIP AT BERTH SHOULD BE CLEARLY DEFINED according to AIS use (Automatic Identification System – AIS). Fig. 1 (left side and right side) should be much bigger where all things are readably.

6.    The Equation (1) should to describe one or more phase(s) regarding ship operating phases in port (waiting time, service time and berthing/unberthing time or some combination between them).

7.    The results which are based on waiting and service time of ships in port including ratio among them may be very appropriate for this kind of study.

8.    Number of ship arrivals (in no. of ships, Table 3) should be changed in average arrival rate. If you know the average service time of any ships at berth and the number of berths, it is easy to calculate the average utilization rate of CTs in a port.

9.    Abbreviations (such as AIS, LA/LB) should be defined at first mention in each of the following sections in the manuscript (title, abstract, text, each figure/table, legend, etc.).

10.              How AIS is used should be clearly and precisely explained in the manuscript.

11.              Tables 4 and 5 should point out which CTs are included in analysis, like in Tables 1-3.

12.              At the end of the Introduction should be included the structure of the manuscript.

13.              The new Conclusions should be written after rewriting the whole manuscript in depth.

Conclusions

This manuscript has potential to be a candidate for publication after significant improvements. It is obvious that a major revision should be an adequate requirement for the manuscript.

Author Response

We appreciate your kind review and revised a manuscript as belows;

  1. The title of the manuscript should be changed – one alternative has been proposed.

In this study, the performance of fully automated terminals during the COVID-19 pandemic is compared with the performance of non-fully automated terminals, so referring to answer No. 2 below, I think it will not be necessary to change the title. We will reflect on your comments in connection with limitations of our future research.

  1. CTs automation should be clearly defined including Automated CTs and Semi-automated CTs.

In this study, the terminals to be analyzed were divided into fully automated and non-fully automated, whereby non-fully automated includes all semi-automated and manual methods, as you mentioned. The purpose of this study is to compare fully automated and non-human terminals in consideration of the problems in ports (container terminals) which were caused by human factors in the COVID-19 situation. Therefore, it is reasonable to conduct a comparison and analysis with the criteria presented in this study. As mentioned in the limitations of this study in the final conclusion, I think it is necessary to perform an analysis at the level of automation (or by type), and I will consider more advanced research topics based on your comments. Thank you.

  1. Some very important papers are missing such as (Nam and Ha (https://ascelibrary.org/doi/abs/10.1061/(ASCE)0733-950X(2001)127:3(171); Orive et al. (doi:10.3390/logistics4010003)) among others.

Thank you for your recommendation of useful papers. We quoted them as appropriate in the Manuscript and included them in the reference list.

  1. Keywords of the manuscript should be changed related to ''Study on Comparing Performance of Automated and Semi-Automated Container Terminals''.

Please understand that we did not change the keywords because we also considered it unnecessary to change the title.

  1. THE SERVICE TIME OF SHIP AT BERTH SHOULD BE CLEARLY DEFINED according to AIS use (Automatic Identification System – AIS). Fig. 1 (left side and right side) should be much bigger where all things are readably.

As you said, we used AIS for illustrating how the service time was calculated in Fig. 1. We enlarged both pictures.

  1. The Equation (1) should to describe one or more phase(s) regarding ship operating phases in port (waiting time, service time and berthing/unberthing time or some combination between them).

Terminal performance was analyzed in this study by calculating the berthing time, and as the reviewer said, waiting time, berthing/unberthing time, etc. were not identified. However, we are performing a joint study with UNCTAD and IAPH to analyze the service level of each port by calculating time in the port, waiting time in the anchorage, and moving time through setting harbor limits and anchorage as shown in Fig. 1 (a). This report will be released later, so please understand that it was not included in this study.

  1. The results which are based on waiting and service time of ships in port including ratio among them may be very appropriate for this kind of study.

As mentioned in the answer to No. 6, this will be announced in a project that we are currently working on with UNCTAD and IAPH.

  1. Number of ship arrivals (in no. of ships, Table 3) should be changed in average arrival rate. If you know the average service time of any ships at berth and the number of berths, it is easy to calculate the average utilization rate of CTs in a port.

As you said, if the average service time and number of berths are identified through the Port Community System (PCS), such as Port-MIS in Korea, the average utilization rate (berth occupancy rate) can be readily obtained. However, as I mentioned earlier, this study did not include this information because it only considered berthing time.

  1. Abbreviations (such as AIS, LA/LB) should be defined at first mention in each of the following sections in the manuscript (title, abstract, text, each figure/table, legend, etc.).

Abbreviations were modified in the manuscript (LA/LB, IMO, hr). However, terminal names such as RWG, AMPT, and QQCTU are used in the same manner as proper nouns, so the full names were not used.

  1. How AIS is used should be clearly and precisely explained in the manuscript.

AIS is a system for identifying a ship's location information, and information on how the berthing time was calculated using this location information is included in the manuscript.

  1. Tables 4 and 5 should point out which CTs are included in analysis, like in Tables 1-3.

The values in Tables 4, 5 included those of 6 fully automated terminals and 9 non-fully automated terminals. Pertinent content is shown in the footnotes of Tables 4 and 5.

  1. At the end of the Introduction should be included the structure of the manuscript.

The structure of the manuscript is included.

  1. The new Conclusions should be written after rewriting the whole manuscript in depth.

This study is meaningful in that it distinctly emphasized the practical merits of introducing a fully automated terminal, and empirically confirmed that it is possible to secure the stability of fully automated terminals, which was theoretically overlooked in previous studies.

Reviewer 4 Report

Following are my comments pertaining to the paper titled "Study on Comparing Performance of Fully Automated Container Terminals During Coronavirus Pandemic",

1. The motivation and background associated with the research work need strengthening by highlighting the importance of the current research work.

2. The contribution of the work need to be thoroughly presented at the end of the Introduction section, if possible you can provide a separate sub-section named contribution.

3. Proper connection between two subsequent works  and two subsequent paragraphs are not established properly. Furthermore, research gaps from the past literature also need to be clearly highlighted at the end of the literature review. Furthermore, the literature review needs to be updated with latest publications from reputed journals within maritime logistics focusing on sustainability aspects and ship speed. Following are some examples,

Bunkering policies for a fuel bunker management problem for liner shipping networks, European Journal of Operational Research

Fuel bunker management strategies within sustainable container shipping operation considering disruption and recovery policies, IEEE Transactions on Engineering Management

Hybridizing basic variable neighborhood search with particle swarm optimization for solving sustainable ship routing and bunker management problem, IEEE Transactions on Intelligent Transportation Systems

Multiobjective approach for sustainable ship routing and scheduling with draft restrictions, IEEE Transactions on Engineering Management

A hybrid dynamic berth allocation planning problem with fuel costs considerations for container terminal port using chemical reaction optimization approach, Annals of operation research

Sustainable maritime inventory routing problem with time window constraints, Engineering Applications of Artificial Intelligence

Composite particle algorithm for sustainable integrated dynamic ship routing and scheduling optimization, Computers & Industrial Engineering

4. The results and discussion section requires more useful insights and managerial implications to highlight the relevance of the work from practical perspective.

5. Furthermore, it is essential that you provide contribution to theory by connecting the implications from results with past literature to clearly elaborate the ways current study is adding value to the body of knowledge.

Author Response

We appreciate your kind review and revised a manuscript as belows;

  1. The motivation and background associated with the research work need strengthening by highlighting the importance of the current research work.
  2. The contribution of the work need to be thoroughly presented at the end of the Introduction section, if possible you can provide a separate sub-section named contribution.

Regarding opinions 1 and 2, I supplemented the content by adding information on the importance and contribution of this study according to your comments. In other words, the importance of securing port stability, the role of a fully automated terminal, and the verification through empirical analysis were added to the middle and last parts of the introduction.

  1. Proper connection between two subsequent works  and two subsequent paragraphs are not established properly. Furthermore, research gaps from the past literature also need to be clearly highlighted at the end of the literature review. Furthermore, the literature review needs to be updated with latest publications from reputed journals within maritime logistics focusing on sustainability aspects and ship speed. Following are some examples, Bunkering policies for a fuel bunker management problem for liner shipping networks, European Journal of Operational Research Fuel bunker management strategies within sustainable container shipping operation considering disruption and recovery policies, IEEE Transactions on Engineering Management Hybridizing basic variable neighborhood search with particle swarm optimization for solving sustainable ship routing and bunker management problem, IEEE Transactions on Intelligent Transportation Systems Multiobjective approach for sustainable ship routing and scheduling with draft restrictions, IEEE Transactions on Engineering Management A hybrid dynamic berth allocation planning problem with fuel costs considerations for container terminal port using chemical reaction optimization approach, Annals of operation research Sustainable maritime inventory routing problem with time window constraints, Engineering Applications of Artificial Intelligence Composite particle algorithm for sustainable integrated dynamic ship routing and scheduling optimization, Computers & Industrial Engineering

Regarding opinion No. 3, the ship operation cost reviewed in this study is presented to show the positive economic impact of the berthing time as a result of the study. As only the cost required during berthing was calculated, only the capital cost and operation cost were considered, except for the fuel cost of the ship. Considering these points, I did not include the recommended paper about management of fuel bunkers, ship scheduling, berth allocation, etc. because I thought it would be too much unrelated information. Please understand this. 

  1. The results and discussion section requires more useful insights and managerial implications to highlight the relevance of the work from practical perspective.
  2. Furthermore, it is essential that you provide contribution to theory by connecting the implications from results with past literature to clearly elaborate the ways current study is adding value to the body of knowledge.

Regarding opinions 4 and 5, as the reviewer mentioned, the practical aspects of the introduction of fully automated terminals were added to the conclusion of the manuscript in relation to previous studies. The biggest implication of this study is that it was empirically confirmed that it is possible to secure the stability of a fully automated terminal, which was theoretically overlooked in previous studies.

Round 2

Reviewer 2 Report

All my recommendations were taken care of by the authors. Thank you

Author Response

We really appreciate your kind review.

Reviewer 3 Report

I would like to recommend to author(s) to consider and apply most of the rest of the suggestions and comments from the 1st round review.

Author Response

We appreciate your kind review and revised the manuscript as below;

First, in line with the reviewer’s comment, the equation was modified by decomposing the total time in port into ship operating phases such as berthing time (BT), waiting time (WT), and moving time (MT). The results of the analysis are presented in Table 3.

However, the objective of this study is not to evaluate the service level for each terminal, so we will exclude the calculation of the MT and WT ratio to BT from this study. Furthermore, regarding the average utilization rate, information such as the number and length of berths that are currently in operation should be accurately identified, but since we are examining terminals of several ports, some port information (e.g., Yangshan Phase 4, QQCTN, etc.)* cannot be confirmed objectively so they have been excluded.

* Yangshan Phase 4 is being developed with 7 berths and QQCTN with 6 berths, but the number of berths in operation cannot be confirmed.

Regarding these two matters, further studies will be conducted after confirming additional information accurately through interviews with each terminal, so we hope you can understand the current situation.

In addition, regarding the title change, as mentioned in the previous answer, the purpose of this study is to examine the performance of fully automated container terminals before and after COVID-19. Therefore, if the title is changed, the research purpose must be completely re-established, and previous studies, as well as the ports and terminals targeted for analysis, also need to be re-examined. Therefore, we respectfully believe that it is correct to proceed with the original title according to the purpose of the study and the subject of the analysis. As for the reviewer's opinion on keywords, I think it is right to leave the keywords as they are unless the title changes.

Thank you for your thorough comments on this, and we ask for your understanding that the title cannot be changed.

Reviewer 4 Report

Authors have adequately addressed the comments and now the paper cn be accepted for publication.

Author Response

We really appreciate your kind review.